# CONTEXTUAL VISION TRANSFORMERS FOR ROBUST REPRESENTATION LEARNING

## ABSTRACT

We introduce Contextual Vision Transformers (ContextViT), a method designed to generate robust image representations for datasets experiencing shifts in latent factors across various groups. Derived from the concept of in-context learning, ContextViT incorporates an additional context token to encapsulate group-specific information. This integration allows the model to adjust the image representation in accordance with the group-specific context. Specifically, for a given input image, ContextViT maps images with identical group membership into this context token, which is appended to the input image tokens. Additionally, we introduce a context inference network to predict such tokens on-the-fly, given a batch of samples from the group. This enables ContextViT to adapt to new testing distributions during inference time. We demonstrate the efficacy of ContextViT across a wide range of applications. In supervised fine-tuning, we show that augmenting pre-trained ViTs with our proposed context conditioning mechanism results in consistent improvements in out-of-distribution generalization on iWildCam and FMoW. We also investigate self-supervised representation learning with ContextViT. Our experiments on the Camelyon17 pathology imaging benchmark and the JUMP-CP microscopy imaging benchmark demonstrate that ContextViT excels in learning stable image featurizations amidst distribution shift, consistently outperforming its ViT counterpart.

## 1 INTRODUCTION

In recent years, Vision Transformers (ViTs) have emerged as a powerful tool for image representation learning (Dosovitskiy et al.). However, real-world datasets often exhibit structured variations or shifts in latent factors across different groups. These shifts, which occur when known or unknown factors of variation change during data acquisition, can lead to inaccurate or biased predictions when the model encounters new data. To address this issue, we introduce Contextual Vision Transformers (ContextViT), a novel method that generates robust feature representations for images, effectively handling variation in underlying latent factors.

Transformer-based models can perform test-time generalization by integrating available information through in-context learning (Brown et al., 2020), where input-label pairs are added to the transformer inputs. In this work, we leverage this principle to address distribution shifts at test time. However, direct application of this strategy presents challenges for vision transformers due to the quadratic scaling of image patch tokens with input resolution.

Derived from the principle of in-context learning, we propose ContextViT, a variant of vision transformer that condenses the relevant context into a single token. This *context token* is shared across all images with the same underlying distribution and varies across different distributions. For instance, in medical imaging, this could correspond to learning a single hospital token for each hospital. The context token is appended to the input sequence of the transformer, conditioning the resulting image representation on the context. This approach reduces the need for conditioning the model on extra input examples and enables efficient inference at test time. Despite these advantages, one limitation remains: it does not allow for generalization to unseen hospitals in absence of inferred context tokens.

To overcome this limitation, we introduce a *context inference network* that estimates these tokens *on-the-fly* from the input images. Given a group of images from the same context (e.g., a hospital),

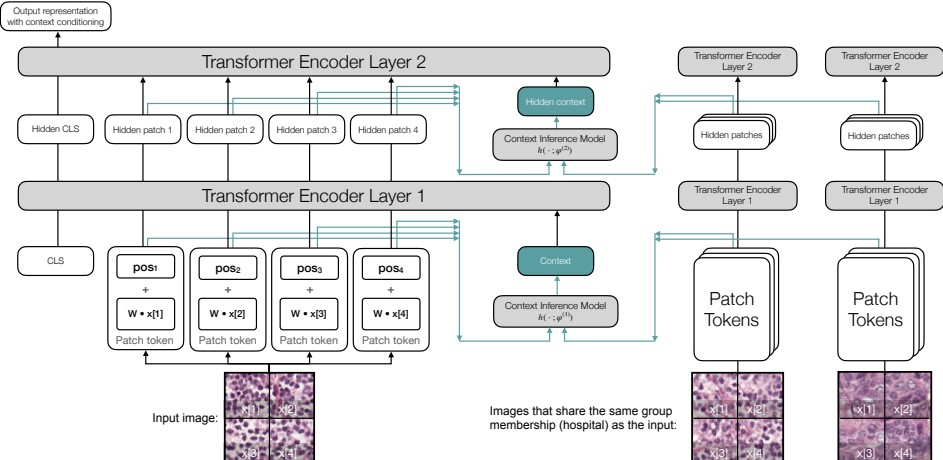

Figure 1: Illustration of ContextViT. For clarity, this figure depicts only two layers of the Transformer. Model components are indicated by grey-colored blocks, while white blocks represent the input, hidden, and output representations. Prior to processing by each Transformer layer, ContextViT employs a context inference model (that aggregates information across all image patches within the current batch sharing identical group membership) to generate the *context token*.

the context inference network predicts the token that encapsulates the characteristics of this context. We argue that this procedure should be applied iteratively across all transformer layers, a process we term *layer-wise context conditioning*. This is because earlier transformer layers capture local image patterns, while later layers capture more abstract concepts Ghiasi et al. (2022). Adjusting the covariates at the first layer is not sufficient to accommodate the covariate shift, which can be a concept-level change.

Our empirical analysis begins with supervised fine-tuning, where we integrate our context conditioning mechanism with pre-trained ViTs. Our results show consistent performance gains across three pre-trained ViT models, namely DINO (Caron et al., 2021), SWAG (Li et al., 2016), and CLIP (Radford et al., 2021), on the WILDS benchmark iWildCam (Beery et al., 2021) and FMoW (Christie et al., 2018). We then explore the integration of self-supervised representation learning with ContextViT. On the microscopy cell imaging benchmark JUMP-CP Haghighi et al. (2022), we observe an interesting phenomenon: while the performance of ViT decreases due to the batch effect as we introduce more data plates for pre-training, the performance of ContextViT steadily increases with more data. On the histopathology benchmark Camelyon17-WILDS (Bandi et al., 2018; Sagawa et al.), ContextViT significantly improves the performance on unseen hospitals, setting a new state-of-the-art.

The main contributions of this paper are:

1. We propose ContextViT, an adaptation of ViT for producing robust image representations using learned context tokens to capture distribution shift. We derive ContextViT from in-context learning under distribution shift.

2. We incorporate a context inference mechanism into ContextViT, enhancing its ability to adapt and generalize to previously unseen distributions on the fly.

3. We present layer-wise context conditioning for ContextViT, a process that uses per-layer context tokens iteratively across all transformer layers to handle concept-level changes across distributions.

4. We explore the integration of self-supervised learning with cell-imaging and histopathology datasets, demonstrating significant performance improvements under distribution shifts and establishing a new state-of-the-art.

## 2 VISION TRANSFORMERS (VITS)

**Patch token embeddings**   Let $x \in \mathbb{R}^{H \times W \times C}$ be an image with $C$ channels and resolution $H$ by $W$. ViTs first partition the image into a sequence of non-overlapping 2D patches $[x_{p_1}, \ldots, x_{p_N}]$,

each with resolution $(H_p, W_p)$ and represented by $x_{p_i} \in \mathbb{R}^{H_p \times W_p \times C}$. ViTs treat each image patch as a "1D token" for the Transformer, and we obtain patch token embeddings by flattening each patch $x_{p_i}$ into a 1D vector and applying a trainable affine projection. The resulting patch token sequence is denoted as $[t_{p_1}, \ldots, t_{p_N}]$ where each $t_{p_i} \in \mathbb{R}^d$.

**CLS token and position embeddings**    In ViTs, we prepend a trainable CLS token $t_{\text{CLS}}$ to the input sequence. This enables the Transformer encoder to capture global image features by aggregating information across the patch sequence. We retain positional information for each patch token using a trainable 1D position embedding $p_i$. The input sequence for the Transformer encoder is thus given by $[t_{\text{CLS}}, t_{p_1} + pos_{p_1}, \ldots, t_{p_N} + pos_{p_N}]$.

**Transformer**    The Transformer layer (Vaswani et al., 2017) is a critical component of the ViT architecture. It comprises a self-attention layer and a feed-forward layer, each with residual connections (He et al., 2016) and layer normalization (Ba et al., 2016). Self-attention enables the model to capture dependencies between patches in the input image by embedding each patch based on its similarity to other patches. ViTs utilize a stack of Transformer layers $T^{(1)}, \ldots T^{(L)}$ to encode the input sequence into a sequence of 1D features:

$$[y_{\text{CLS}}^{(L)}, y_{p_1}^{(L)}, \ldots, y_{p_N}^{(L)}] = T^{(L)} \cdots T^{(2)} T^{(1)}([t_{\text{CLS}}, t_{p_1} + pos_{p_1}, \ldots, t_{p_N} + pos_{p_N}]).$$

## 3  CONTEXTUAL VISION TRANSFORMER (CONTEXTVIT)

Let $\mathcal{D}_c$ denote a collection of samples from a distribution $P_c(\cdot)$ under a context with index $c$, which can point to an unobserved environmental effect, a set of covariates such as experimental conditions, or another factor of variation with a scope over the set $\mathcal{D}_c$ that systematically shapes its distribution. Our goal is as follows: given a collection of such related datasets from distributions with varying factors of variation, captured as varying contexts $c$, we would like to learn a model that can generalize gracefully across these different domains and ideally adapt to new contexts $c^*$ during test time.

We assume that we observe data from multiple distributions jointly, each varying by a context $c$, so that our dataset is comprised by their collection $\mathcal{D} = \{\mathcal{D}_1, ..., \mathcal{D}_c\}$. We aim to learn a single shared ViT model, parameterized by $\theta$, across all sub-datasets with varying contexts. We denote a tuple $\{x, y, c\}$ as the data describing the sample, where $x$ denotes an input, the image, and $y$ denotes an output variable such as a label or an unknown embedding, and $c$ denoting *group membership* for each datum to each sub-distribution $P_c(\cdot)$. Over the next sections we will explain our assumptions for ContextViT and will present practical implementations.

### 3.1  IN-CONTEXT MODEL

A popular paradigm for incorporating test-time conditioning in Transformers is given by in-context learning (Brown et al., 2020) and prompt-conditioning (Radford et al., 2021). To translate that paradigm to our task, we incorporate group membership to the representation learning task by conditioning on the members of the group, assuming the following factorization of the joint distribution over all data:

$$P(Y|X, C) = \prod_c \prod_{\{x,y\} \in \mathcal{D}_c} P(y|x, \mathcal{D}_c; \theta). \tag{1}$$

Here, we would concatenate all images belonging to context $c$ to the query image $x$. During ViT processing, these images are patchified and represented as patch tokens, resulting in an input to the Transformer consisting of the tokens belonging to the query image and to the context images. This process is closest to in-context reasoning as commonly applied for LLMs, but ported over to the scenario of representation learning for vision.[1]

---

[1]In our formulation, we exclusively utilize the data $x$ as the context, rather than the data-label pair $(x, y$. This decision is informed by the practical reality that, while test distribution labels may be scarce or entirely absent, data itself is typically abundant.

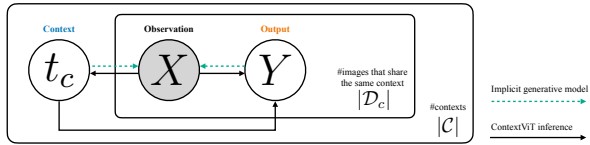

Figure 2: ContextViT inference (solid line) vs. the implicit generative model (dotted line)

## 3.2 CONTEXT-TOKEN MODEL

For the sake of exposition, let's assume an implicit hierarchical generative model of images with shared global latent variables per group $t_c \in \mathbb{R}^d$ which summarizes the shared latent characteristics of $\mathcal{D}_c$. This variable represents an embedding of the context (which can characterize the environment or implicit covariates determining each distribution, or other factors of variation shifting in $c$) and is treated as a *context token*, see Fig. 2. Such models are common in generative modeling when environmental effects or style-class disentanglement are proposed. The context token $t_c$ can be characterized by its posterior distribution given the members of the group $P(t_c|\mathcal{D}_c)$ and can then be utilized in an adjusted model $P(y|x, t_c; \theta)$ to explain the impact of $\mathcal{D}_c$.

Upon closer inspection, it is evident that the in-context-learning process described in Sec. 3.1 can be interpreted as inferring and marginalizing the posterior distribution $P(t_c|\mathcal{D}_c)$ of the context-specific variable $t_c$ as shown in the following:

$$P(y|x, \mathcal{D}_c) = \int P(y|x, t_c; \theta) P(t_c|\mathcal{D}_c) \, \mathrm{d}t_c. \tag{2}$$

Under this viewpoint using Eq.2, we can now see that the context token is shared over all members of a distribution with index $c$ (similar to a variable shared in a plate in graphical models), allowing us to rewrite Eq.1 such that we can factorize the dependence of the model on $\mathcal{D}_c$ for each data point given this context token:

$$P(Y|X, C) = \prod_c \int P(t_c|\mathcal{D}_c) \prod_{\{x,y\} \in \mathcal{D}_c} P(y|x, t_c; \theta) \, \mathrm{d}t_c. \tag{3}$$

We now have established equivalence between the in-context model, and performing inference in an implicit forward generative process which assumes existence of a context token as an additional factor of variation capturing distribution shifts for each group sharing a context. We note that recent work proposes a deeply related interpretation of in-context learning in (Xie et al., 2022) supporting our view of an implicit probabilistic model.

## 3.3 ORACLE-CONTEXT MODEL

To simplify the setup for use in a ViT framework, we devise simpler formulations of the model and perform maximum likelihood inference over the context token, simplifying Eq. 3 to:

$$P(Y|X, C) = \prod_c \prod_{\{x,y\} \in \mathcal{D}_c} P(y|x; t_c, \theta), \tag{4}$$

where $t_c$ now is a shared parameter that can be inferred per existing group during training. We denote this model the Oracle-Context Model, since it only assumes knowledge of the indicator $c$ about which group an image belongs to, and does not require access to the other members of the group.

We can instantiate this oracle model $P(y|x; t_c, \theta)$ by conditioning the ViTs with a learnable context token $t_c$ and append it to the input sequence of the Transformer: $[t_{\mathrm{CLS}}, t_c, t_1 + pos_1, \ldots, t_N + pos_N]$.

The limitation, however, is that such a method cannot be applied during test-time beyond known distributions, since it lacks the ability to infer the token embedding without prior exposure to examples from this group during the training phase.

### 3.4 CONTEXT INFERENCE MODEL

We overcome this limitation by closer matching the objective and assuming a model that performs amortized inference over the context-token parameters on the fly given observations $\mathcal{D}_c$ by utilizing an inference network $h(\cdot; \phi)$. Concretely, we will again simplify $p(t_c|\mathcal{D}_c)$ to be given by maximum likelihood but this time also utilize $\mathcal{D}_c$ by posing:

$$P(Y|X, C) = \prod_c \prod_{\{x,y\} \in \mathcal{D}_c} P(y|x, t_c; \theta), \quad \text{with } t_c = h(\mathcal{D}_c; \phi). \tag{5}$$

Here, $h(\cdot; \phi)$ infers $t_c$ *on the fly* when observing $\mathcal{D}_c$, and can indeed also be used during test time on a previous *unknown distribution* with context $c^*$ time given a set of samples $\mathcal{D}_{c^*}$. We note that the assumption of availability of samples $\mathcal{D}_c$ during testing is highly realistic and common, as datasets $\mathcal{D}_c$ (comprised of observed inputs sharing a context) typically appear in groups for free also during test time, for example one would never image a single cell in an experiment but a collection under fixed conditions, and the expensive step is to assess their unknown mapping to output labels per group. In our experiments we will show how contexts $c$ can also be persistent objects, such as hospitals having measurement processes for pathology images, that easily allow us to access samples from the set of measurements given context $c$.

There are multiple options available for instantiating the inference model $h(\cdot; \phi)$ (refer to the Appendix for more details). In the case of ContextViT (Figure 1), the inference model $h(\cdot; \phi)$ comprises three key components:

**Mean pooling**     Given an image $x \in \mathcal{D}_c$ with its corresponding patch embeddings $t_{p_i}$, a direct way to obtain the context token $t_c$ is to aggregate the patch embeddings of all images in $\mathcal{D}_c$ through an average pooling operation $t_c^{\text{mean}} = h_c^{\text{mean}}(\mathcal{D}_c) \coloneqq \frac{1}{|\mathcal{D}_c|} \sum_{x \in \mathcal{D}_c} \frac{1}{N} \sum_{x_{p_i} \in x} t_{p_i}$. The input sequence for the image $x$ would then be represented as $[t_{\text{CLS}}, t_c^{\text{mean}}, t_1 + pos_1, \ldots, t_N + pos_N]$. In practice, instead of pooling over the entire $\mathcal{D}_c$, we apply mean pooling over $\mathcal{B} \cap \mathcal{D}_c$ where $\mathcal{B}$ is the current batch.

**Linear transformation with gradients detaching**     To further enhance the context representation, we introduce a trainable linear transformation to the pooled representation. In order to prevent the patch embeddings from being penalized by the context inference model, we detach their gradients. This results in the expression $h_c^{\text{linear}}(\mathcal{D}_c; b, W) \coloneqq b + W \cdot \texttt{detach}(t_c^{\text{mean}})$.

**Layerwise context conditioning**     Recent work (Ghiasi et al., 2022) has shown that Transformer features progress from abstract patterns in early layers to concrete objects in later layers. We explore the application of context conditioning beyond the input layer driven by the hypothesis that patch embeddings may not be able to capture higher-level concepts. For the $l$-th Transformer layer, we use $y^{(l)}$ to denote its output and $\mathcal{D}_c^{(l)}$ to denote the collection of *hidden* patch embeddings of context $c$. We propose *layerwise* context conditioning which performs amortized inference over context-token parameters for each layer in the ViT instead of just propagating the input-layer token:

$$P(y^{(l)}|y^{(0)} \coloneqq x; t_c) = \prod_{l=1}^{L} P(y^{(l)}|y^{(l-1)}; t_c^{(l-1)}), \quad \text{with } t_c^{(l)} = h(\mathcal{D}_c^{(l)}; \phi^{(l)}).$$

For the $l$-th Transformer layer, we can express the layerwise context conditioning as

$$[y_{\text{CLS}}^{(l)}, y_c^{(l)}, y_{p_1}^{(l)}, \ldots, y_{p_N}^{(l)}] = T^{(l)}([y_{\text{CLS}}^{(l-1)}, t_c^{(l-1)}, y_{p_1}^{(l-1)}, \ldots, y_{p_N}^{(l-1)}]).$$

## 4 EXPERIMENTS

We demonstrate the utilities of ContextViT across a variety of applications. We start with the common supervised fine-tuning setting where we augment three existing pre-trained ViTs with our context inference model (Section 4.1). Next we experiment ContextViT with self-supervised representation learning. We demonstrate the importance of context conditioning for both in-distribution generalization 4.2 and out-of-distribution generalization 4.3. We present our ablation study and analysis in

Table 1: OOD accuracy on iWIldCam and worst-region OOD accuracy on FMoW for supervised fine-tuning (FT). Augmenting existing pre-trained ViTs with our context inference model consistently improves the out-of-distribution generalization performance. Previous work: FLYP (Goyal et al., 2022), Model Soups (Wortsman et al., 2022), FT ViT (Wortsman et al., 2022; Kumar et al., 2022).

| | iWildCam | | | FMoW | | |
|---|---|---|---|---|---|---|
| | DINO ViT-S/8 | SWAG ViT-B/16 | CLIP ViT-L/14 | DINO ViT-S/8 | SWAG ViT-B/16 | CLIP ViT-L/14 |
| FLYP | — | — | $76.2_{\pm0.4}$ | — | — | — |
| Model Soups | — | — | $79.3_{\pm0.3}$ | — | — | $47.6_{\pm0.3}$ |
| FT ViT | — | — | $78.3_{\pm1.1}$ | — | — | $49.9$ |
| *Our implementations* | | | | | | |
| FT ViT | $75.7_{\pm0.2}$ | $77.5_{\pm0.5}$ | $81.5_{\pm0.2}$ | $35.0_{\pm0.2}$ | $39.8_{\pm0.5}$ | $46.6_{\pm1.1}$ |
| FT ContextViT (w/o layerwise) | $77.5_{\pm0.2}$ | $78.6_{\pm0.4}$ | $81.9_{\pm0.1}$ | $37.3_{\pm0.6}$ | $41.3_{\pm1.0}$ | $49.1_{\pm0.7}$ |
| FT ContextViT | $\mathbf{77.7}_{\pm0.3}$ | $\mathbf{79.6}_{\pm0.6}$ | $\mathbf{82.9}_{\pm0.3}$ | $\mathbf{37.7}_{\pm0.5}$ | $\mathbf{41.4}_{\pm0.3}$ | $\mathbf{49.9}_{\pm0.4}$ |

Section 4.3. **Due to the constraints of space, we kindly direct our readers to the Appendix for the pseudo code implementation details. We are committed to release our code.**

## 4.1 FINE-TUNING PRE-TRAINED VITS WITH CONTEXT CONDITIONING

To evaluate robustness over data shifts, we consider two image classification datasets from the WILDS benchmarks (Koh et al., 2021): iWildCam (Beery et al., 2021) and FMoW (Christie et al., 2018). In iWildCam, the task is to classify the animal species (182-way classification) from images taken by different cameeras. We use the camera trap id for context inference and report the testing accuracy on unseen camera traps. In FMoW, the task is to classify the land use of a satellite image (62-way classification). We use the region id for context inference and report the worst-region accuracy for image from the testing time period.

We consider three existing pre-trained ViT models, each with a different number of parameters: 1) DINO ViT-S/8 (Caron et al., 2021), which is pre-trained on ImageNet with self-distillation between a teacher and a student network. 2) SWAG ViT-B/16 (Singh et al., 2022), which is pre-trained on IG3.6B using weak supervision (hashtags from Instagram). 3) CLIP ViT-L/14, which is pre-trained on the Multi-modal WebImageText using language-image contrastive learning. Despite their differences in pre-training objectives and datasets, these models share the same ViT backbone. *Therefore we can augment these pre-trained models with our proposed context conditioning mechanism and fine-tune the combined ContextViT jointly with empirical risk minimization.*

Table 1 presents our results for supervised fine-tuning. We note that our direct fine-tuning baseline (using CLIP ViT-L/14) outperforms the number reported by Wortsman et al. (2022) on iWildCam (81.5 vs. 78.3) and underperforms the number reported by Kumar et al. (2022) on FMoW (46.6 vs. 49.9). We think this is likely caused by the differences in the implementation details (data augmentation, learning rate scheduling, etc.), and unfortunately we cannot find the configuration online to reproduce the exact numbers. Nevertheless, upon comparing our implementations, we make the following observations: 1) Smaller ViTs exhibit inferior generalization compared to larger ViTs; 2) Incorporating our context inference model consistently enhances the performance of ViTs; 3) Layerwise context inference further enhances generalization.

## 4.2 IN-DISTRIBUTION GENERALIZATION WITH SELF-SUPERVISED LEARNING

Microscopy imaging with cell painting has demonstrated its effectiveness in studying the effects of cellular perturbations (Haghighi et al., 2022; Moshkov et al., 2022; Sivanandan et al.). Despite meticulous regulation of experimental parameters like cell density and exposure time, technical artifacts can still confound measurements from these screens across different batches. Learning a robust cell representation that can effectively handle batch variations remains an ongoing challenge.

**Dataset** We consider three cell plates (`BR00116991`, `BR00116993`, `BR00117000`) from the JUMP-CP dataset released by the JUMP-Cell Painting Consortium (Chandrasekaran et al., 2022). Each plate consists of 384 wells with perturbations (either via a chemical compound or a crispr

Table 2: Accuracy of 160-way gene perturbation classification for three cell plates in JUMP-CP. For each plate, we train a classifier (one-layer MLP) on top of the pre-trained DINO embeddings and evaluate its held out performance. The DINO embeddings are pre-trained on either a single plate `BR00116991` (first section) or all three plates combined (second section).

| Method | BR00116991 | BR00116993 | BR00117000 |
|---|---|---|---|
| *DINO pre-training on BR00116991 only* | | | |
| ViT-S/16 | 56.5 | 42.7 | 46.8 |
| *DINO pre-training on BR00116991 & BR00116993 & BR00117000* | | | |
| ViT-S/16 | 53.6 | 43.5 | 45.6 |
| ContextViT-S/16 | **57.7** | **48.0** | **48.8** |

guide) targeted at 160 different genes. Our input data consists of single-cell images: 229228 images for `BR00116991`, 226311 images for `BR00116993` and 239347 images for `BR00117000`. We note that here each single-cell image has dimension $224 \times 224 \times 8$ (5 fluorescence channels and 3 brightfield channels). For each plate, we split the images randomly into training (40%), validation (10%) and testing (50%). We use the plate id for context inference.

**Results** For each model, we use DINO (Caron et al., 2021) to pre-train the ViT and ContextViT from scratch for 100 epochs. Once pre-training is completed, we freeze the backbone parameters and attach an MLP (with one hidden ReLU layer of dimension 384) to predict the targeted gene of the cell. Given the unique nature of each plate, we train an individual MLP classifier for **each separate plate** using the training split of that specific plate, and subsequently report the corresponding testing accuracy in Table 2.

We note that the representation learned by DINO on a single plate (`BR00116991`) exhibits poor generalization across plates (42.7 for `BR00116993` and 46.8 for `BR00117000`). Moreover, directly combining all three plates for pre-training results in a degradation of the model's performance: -2.9 for `BR00116991`, +0.8 for `BR00116993` and -1.2 for `BR00117000`. In contrast, by utilizing ContextViT, the model effectively accounts for batch effects through context conditioning during the pre-training stage, resulting in superior performance across all three plates.

### 4.3 OUT-OF-DISTRIBUTION GENERALIZATION WITH SELF-SUPERVISED LEARNING

In medical applications, models are often trained using data from a limited number of hospitals with the intention of deploying them across other hospitals more broadly (Yala et al., 2019; 2021; Bandi et al., 2018). However, this presents a challenge for the generalization of out-of-distribution data. In this section, we aim to evaluate the ability of self-supervised learning with ContextViT to achieve better out-of-distribution generalization.

**Dataset** We consider the Camelyon17-WILDS benchmark (Bandi et al., 2018; Sagawa et al.). The dataset contains 455,954 labeled and 2,999,307 unlabeled pathology images across five hospitals (3 for training, 1 for validation and 1 for testing). Given a $96 \times 96 \times 3$ image, the task is to predict whether the image contains any tumor tissue. *We use the hospital id for context inference.*

**Results** We utilize DINO to pre-train our ViT and ContextViT models from scratch on unlabeled pathology images and evaluate them through *linear probing* Chen et al. (2020). Unlike previous work ((Sagawa et al., 2021)) that incorporates both in-distribution and out-of-distribution unlabeled data (Unlabeled ID&OOD), *we exclusively use images from the three training hospitals, denoted as "Unlabeled ID", during pre-training stage to prevent any potential information leakage from out-of-distribution data.* Given the 96 by 96 input resolution of our dataset, we opt for a smaller patch size (8 instead of 16) for the ViT models. Once pre-training is complete, we freeze the ViT parameters and train a linear classifier, **shared across all training hospitals**, with SGD to predict the target label, the presence of tumors. We report the testing accuracy on the out-of-distribution hospitals. Intuitively, if the model is capable of extracting representations that remain consistent across different hospitals, the linear classifier would exhibit strong performance on the unseen hospitals.

Table 3 presents a comparison of our linear probing results (marked with[†]) with other published results on the Camelyon17 benchmark. Our DINO pre-trained ViT-S/8 baseline outperforms the much

Table 3: Accuracy of linear probing of pre-trained DINO embeddings on Camelyon17-WILDS. We use [†] to denote our implementations. Unlabeled ID&OOD denotes unlabeled pathology images from all hospitals, while Unlabeled ID denotes unlabeled images from the three training hospitals. DenseNet121 results are adopted from Koh et al. (2021); Sagawa et al. (2021). The results of ViT-B/16 and ViT-L/14 are adopted from Kumar et al. (2022).

| Backbone | Size | Pre-training Method | Pre-training Dataset | In-distribution Accuracy | OOD Accuracy |
|---|---|---|---|---|---|
| *Fine-tuning all parameters* | | | | | |
| DenseNet121 | 7.9M | Supervised | ImageNet1K | 90.6 | 82.0 |
| DenseNet121 | 7.9M | SwaV | Unlabeled ID&OOD | 92.3 | 91.4 |
| ViT-B/16 | 86M | DINO | ImageNet | — | 90.6 |
| ViT-L/14 | 303M | CLIP | WebImageText | 95.2 | 96.5 |
| *Linear probing on frozen backbone* | | | | | |
| ViT-L/14 | 303M | CLIP | WebImageText | — | 92.6 |
| ViT-S/8[†] | 21.7M | DINO | Unlabeled ID | **98.9** | 93.8 |
| ContextViT-S/8[†] | 21.8M | DINO | Unlabeled ID | **98.9** | **97.5** |

Table 4: Ablation study and pre-training time cost (on a server with 8 A100 GPUs) of different context inference models. For each method, we pre-train the corresponding ContextViT on Camelyon17-WILDS (Unlabeled ID) from scratch using DINO for 100 epochs. We report the out-of-distribution accuracy of linear probing on the official OOD testing split.

| Context Inference Method | Linear Probing OOD Acc. | DINO Pre-training Time Cost |
|---|---|---|
| No context (ViT-S/8 baseline) | 93.8 | **21.5h** |
| Mean patch embeddings | 94.1 | 22.6h |
| Mean patch embeddings + linear | 94.4 | 23.5h |
| Mean detached patch embeddings + linear | 97.2 | 23.2h |
| Layerwise mean detached patch embeddings + linear | **97.5** | 29.9h |
| Deep sets over patch embeddings | 92.5 | 30.0h |
| Deep sets over detached patch embeddings | 94.9 | 29.1h |

larger CLIP model in terms of linear probing (93.8 vs. 92.6), which is not surprising given the ViT-S/8 has been pre-trained on the same pathology domain. Next, we see that by conditioning our ViT-S/8 representation with other images from the same hospital within the testing batch, ContextViT-S/8 achieves an OOD accuracy of 97.5%, significantly outperforming all baselines. Moreover, Figure 4 demonstrates that the linear classifier built upon ContextViT-S/8 continues to enhance its OOD performance as the training of the linear classifier progresses, while the one built upon ViT-S/8 exhibits signs of over-fitting to the training data.

**Ablation study on the context inference model** Due to space constraints, we have deferred additional analyses to the Appendix. In Table 4, we perform an ablation study of our context inference model $h(\cdot; \phi)$. We also experimented with other set representation models for context inference, including deep sets (Zaheer et al., 2017) (see Appendix for details). We observe that utilizing the mean patch embeddings as context (94.1) and incorporating an additional linear transformation (94.4) result in improved performance compared to the no context baseline (93.8). Notably, we find that detaching the gradients when calculating the context token from the patch embeddings is crucial, leading to a performance boost from 94.4 to 97.2. Furthermore, the application of layerwise context conditioning further enhance the performance.

**Limitations** One limitation of ContextViT is the extra time cost for the context conditioning. In Table 4, we present the DINO pre-training time cost for different context inference methods. We observe that adding the non-layerwise context conditioning increases the baseline DINO pre-training time by 5%-9%. Applying layerwise context conditioning further increases the time cost by 29%.

ContextViT excels in imaging applications where data naturally segregate into distinct groups or batches. However, its efficacy is contingent upon the accurate specification of group membership variables that genuinely reflect the environmental context of the data (Figure 2). Should these variables be mis-specified—failing to encapsulate the true environmental factors—ContextViT may not outperform a standard ViT. For instance, a context variable assigned arbitrarily would not contribute to the model's generalization power. Therefore, the success of ContextViT is intimately tied to the user's insight into the dataset.

## 5 RELATED WORK

**In-context learning** Our work draws inspiration from and expands the idea of in-context learning (Brown et al., 2020). We realize a form of conditioning that can position transformer-based models to perform test-time adaptation to new tasks, but instead of conditioning on a dataset explicitly for each query, we infer context tokens that are shareable across a group. Like Xie et al. (2022), we interpret in-context learning via an implicit generative model. The key distinction lies in our focus on adaptation to distribution shift, and our expansion of the framework to include explicit context tokens.

There is a substantial body of previous work on conditioning transformers with extra tokens. Li and Liang (2021) introduced prefix tuning as a method to adapt a frozen Transformer model for various downstream tasks. Xu et al. (2022); Li et al. (2021); Mao et al. (2022a) employed additional tokens to encode extra domain knowledge, such as multi-modal information from language. We focus on the scenario where we do not assume existence multi-modal measurements and our domain knowledge is structural (knowing group membership). White et al. (2022) explored hierarchical models for transformers in the context of language modeling, highlighting similar group-specific effects and connections to mixed models. Zhang et al. (2023); Jia et al. (2022); Zhou et al. (2022) learns extra prompt tokens for different downstream imaging applications. ContextViT distinguishes itself by introducing the conditioning mechanism *based on the group membership of the data*. Unlike previous approaches, *ContextViT derives context information directly from the input image and applies context conditioning at every layer of the transformer encoder.* This enables ContextViT to effectively generalize across unseen data groups.

**Robust learning** Robustness in machine learning can be interpreted in various ways. Mahmood et al. (2021) discovered that ViTs are vulnerable to white-box adversarial attacks Hendrycks and Dietterich (2019). Consequently, numerous strategies have been proposed to enhance the adversarial robustness of ViTs ViTs (Mao et al., 2022b; Chefer et al., 2022).

In this study, we concentrate on robustness in the face of systematic distribution shifts. Test time adaptation is a widely adopted approach (Sun et al., 2020; Liu et al., 2021; Zhang et al., 2022). Li et al. (2022) developed a matching network to select the most suitable pretrained models for testing. The use of batch normalization at test time has also been shown to improve the generalization of CNN across various applications (Zhang et al., 2021; Nado et al., 2020; Li et al., 2016; Schneider et al., 2020; Lin and Lu, 2022; Kaku et al., 2020). Our approach, however, diverges from directly normalizing feature statistics or training multiple expert models. Instead, we incorporate context information as an additional latent input to the Transformer layer. Finally, it's worth noting that the representations generated by ContextViT can be seamlessly integrated with robust learning algorithms such as Group Distributionally Robust Optimization (Sagawa et al., 2019) and Invariant Risk Minimization (Arjovsky et al., 2020), should users desire a specific form of invariance.

## 6 CONCLUSION

We have presented Contextual Vision Transformers, a method that addresses challenges posed by structured variations and covariate shifts in image datasets. By leveraging the concept of in-context learning and introducing context tokens and token inference models, ContextViT enables robust feature representation learning across groups with shared characteristics. Through extensive experimental evaluations across diverse applications, ContextViT consistently demonstrates its utility compared to standard ViT models in terms of out-of-distribution generalization and resilience to batch effects. This success highlights the power and versatility of ContextViT when handling structured variations in real-world applications, where invariance to such nuisances may be desirable.

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

APPENDIX

## A   IMPLEMENTATION DETAILS ON SUPERVISED FINE-TUNING

### A.1   PSEUDO CODE

The integration of ContextViT with existing pre-trained ViT models is straightforward. As illustrated in Figure 3, we provide a PyTorch-style pseudo code to outline the process.

```python
1  # self.cls_token            The CLS token [1, 1, embed_dim]
2  # self.pos_embed            The positional embeddings [1, seq_len, embed_dim]
3  # self.transformer_layers   Sequence (nn.Sequential) of transformer layers.
4  # self.linear_layers        List (nn.ModuleList) of linear layers.
5
6  def forward(self, x, c):
7      # Forward pass of ContextViT
8      # Arguments:
9      #   x     images of dimension: [bs, n_channels, H, W]
10     #   c     covariates of x with dimension: [bs]
11     # Output:
12     #   y     image embeddings: [bs, embed_dim]
13
14     # Convert input images into sequences of patches
15     x = self.patch_embed(x)  # [bs, seq_len, embed_dim]
16
17     # Apply the context inference model to compute the context token
18     context_token = self.context_inference(x, c, layer_id=0)
19     # [bs, 1, embed_dim]
20
21     # concatenate the CLS token, context token and the patch tokens
22     x = torch.cat([self.cls_token, context_token, x+self.pos_embed], dim=1)
23     # [bs, seq_len+2, embed_dim]
24
25     for layer_id, transformer in enumerate(self.transformer_layers):
26         # Apply one transformer layer
27         x = transformer(x)  # [bs, seq_len+2, embed_dim]
28
29         # Compute the context token from the hidden patch embeddings
30         x[:,1:2] = self.context_inference(x[:,2:], c, layer=layer_id+1)
31
32     # return the image embedding at the cls token
33     y = self.norm(x)[:, 0]
34     return y
35
36
37  def context_inference(self, x, c, layer_id):
38      # Context inference model
39      # Arguments:
40      #   x          sequences of patch embeddings: [bs, seq_len, embed_dim]
41      #   c          covariates of x with dimension: [bs]
42      #   layer_id   index of the transformer layer: int
43      # Output:
44      #   context    context token: [bs, 1, embed_dim]
45
46      # Group patches with the same covariate value
47      unique, inverse = torch.unique(c, return_inverse=True)
48
49      # Initialize context token
50      context = torch.zeros([bs, 1, embed_dim])
51
52      # Infer context token over examples with the same covariate value
53      for idx, u in enumerate(unique):
54          # Detached mean pooling over patches that have covariate value u
55          m = torch.mean(x[inverse == idx].detach(), dim=[0, 1])
56
57          # Apply a linear layer (based on the transformer layer id)
58          context[c==u, 0] = self.linear_layers[layer_id](m)
59
60      return context
```

Figure 3: PyTorch-style pseudo code for ContextViT.

## A.2  DATASET DETAILS

To evaluate robustness over data shifts, we consider two image classification datasets from WILDS (Koh et al., 2021): iWildCam (Beery et al., 2021) and FMoW (Christie et al., 2018).

iWildCam consists of 203,029 labeled images captured different camera traps. The task is to classify the animal species among 182 possible options. We use the official data split. At test time, we measure the *average classification accuracy* on unseen camera traps to assess out-of-distribution generalization. We use the camera trap id for the context inference.

FMoW includes 141,696 satellite images labeled for 62 different land use categories, such as shopping malls. The images are also tagged with the year they were taken and their geographical region. We adopt the standard split, training our model on images from 2002-2013 and validating it on images from 2013-2016. We evaluate the model's *worst-region accuracy* on out-of-distribution images (from 2016-2018) during testing. We use the region id for the context inference.

## A.3  DATA AUGMENTATION

We apply standard data augmentations during supervised fine-tuning for both ContextViTs and ViTs. These augmentations include random resized cropping, random horizontal flipping, and random color jittering.

For the iWildcam dataset, we set the crop scale to be [0.08, 1.0], the horizontal flip probability is set to 0.5, and color jitter is applied with a maximum brightness change of 0.4, maximum contrast change of 0.4, maximum saturation change of 0.4, and maximum hue change of 0.1, with a probability of 0.8.

In the FMoW dataset, objects typically only appear in a small part of the image, so aggressive cropping may introduce noise. Therefore, we set the crop scale to be [0.8, 1.0] and reduce the probability of color jittering to 0.1.

## A.4  OPTIMIZATION

We conduct fine-tuning of both ViTs and ContextViTs with a batch size of 256 for a total of 20 epochs. For optimization, we follow previous work Wortsman et al. (2022) and utilize the AdamW optimizer Loshchilov and Hutter (2019). To ensure a stable training process, we perform a two-epoch warm-up for the optimizer before applying a cosine annealing scheduler to decay the learning rate. We tune the learning rate within the range of $[10^{-4}, 10^{-5}, 10^{-6}]$. Weight decay is applied to the model, excluding the bias parameters. Inspired by Caron et al. (2021), we set the weight decay to 0.04 at the beginning and gradually increase it using a cosine scheduler to reach 0.4 at the end of training. We fine-tune each model for 20 epochs with a batch size of 256. We perform model selection based on the accuracy on the official validation split and report the mean and standard deviation across five runs.

# B  SELF-SUPERVISED LEARNING ON MICROSCOPY CELL IMAGING: JUMP-CP

## B.1  DINO PRE-TRAINING: DATA AUGMENTATION

Each single-cell image in the JUMP-CP dataset has dimensions of 224 by 224 by 8. Since the images are consistently captured at a fixed magnification ratio, we have refrained from applying random scaling, as it could hinder the model's ability to infer the absolute size of the cells accurately. Instead, we have employed the following data augmentations: random padded cropping, horizontal and vertical flipping, rotation (with angles of 90, 180, and 270 degrees), defocus Hendrycks and Dietterich, coarse dropout DeVries and Taylor (2017), and input channel dropout Tompson et al. (2015). To facilitate these augmentations, we have utilized the Albumentations package Buslaev et al. (2020), which supports an arbitrary number of channels.

During DINO pre-training, the teacher network receives two global views of the image, while the student network receives six local views. Here, we provide an explanation of the data augmentation configurations for both the global and local views.

For the teacher network in DINO, the two global views are generated as follows: starting with the original single-cell image, we first apply random padding to expand the image to $256 \times 256$ and subsequently crop it to $224 \times 224$. We apply flipping and rotating transformations uniformly at random. In one view, we apply defocus with a radius range of $(1, 3)$, while in the other view, the defocus radius range is $(1, 5)$. Additionally, we apply coarse dropout, allowing for a maximum of 10 cutout holes, where each hole has a maximum dimension of 10 by 10. However, we do not apply input channel dropout for the global views.

For the student network, which receives eight local views, we follow a similar process as with the global views. Starting with the original image, we apply random padding to expand it to $256 \times 256$ and then crop it to $96 \times 96$. We apply flipping and rotating transformations uniformly at random, and we use the same defocus radius range of $(1, 3)$ for all six local views. Instead of coarse dropout, we randomly dropout the input channels with a probability of 0.2.

### B.2 DINO PRE-TRAINING: OPTIMIZATION

Our optimization configuration for the DINO pre-training stage closely follows the guidelines provided in the Caron et al. (2021) GitHub repository. We utilize a batch size of 512 and train DINO for a total of 100 epochs. The key aspects of our configuration are as follows:

- We employ the AdamW optimizer Loshchilov and Hutter (2019) and initiate the learning rate warm-up phase for the first 10 epochs. Considering our batch size, we set the maximum learning rate to 0.001, following recommendations from You et al. (2018); Caron et al. (2021). Subsequently, we decay the learning rate using a cosine learning rate scheduler. Our target learning rate at the end of optimization is set to $10^{-6}$.

- Weight decay is applied to all parameters except for the biases. We set the initial weight decay to 0.04 and gradually increase it to 0.4 using a cosine learning rate scheduler towards the end of training.

- The DINO projection head we utilize has 65536 dimensions, and we do not employ batch normalization in the projection head.

- The output temperature of the teacher network is initially set to 0.04 and is linearly increased to 0.07 within the first 30 epochs. Throughout the remainder of training, the temperature is maintained at 0.07. Additionally, during the first epoch, we freeze the parameters of the output layer to enhance training stability.

### B.3 DOWNSTREAM CLASSIFIER

Based on our preliminary study on `BR00116991` using ViT-S/16, we found that a multi-layer perceptron (MLP) with two hidden layers outperforms the linear classifier (53.6% accuracy vs. 10.4% accuracy). Our final MLP architecture consists of two hidden layers with ReLU activations, and each hidden layer has a dimension of 512. To optimize the parameters of the MLP, we employ the Adam optimizer Kingma and Ba (2014) with a batch size of 256, and we train the model for a maximum of 100 epochs. A weight decay of $10^{-5}$ is applied, and we fine-tune the learning rate within the range of $\in [10^{-3}, 10^{-4}, 10^{-5}]$ to find the optimal value.

## C SELF-SUPERVISED LEARNING ON HISTOPATHOLOGY: CAMELYON17-WILDS

### C.1 DINO PRE-TRAINING: DATA AUGMENTATION

The Camelyon17-WILDS dataset is a patch-based variant of the Camelyon17 dataset, where each pathology image has a dimension of 96 by 96. To enable finer-grained reasoning in ViTs, we use a smaller patch size of 8 by 8. The images are stored in the RGB pixel format, consisting of 3 bytes per pixel. For pre-training the DINO embeddings, we apply standard data augmentations, as done in previous work Caron et al. (2021). These augmentations include random resizing and cropping, horizontal flipping, random color jittering, random grayscale transformation, Gaussian blur, and solarization.

Table 5: OOD accuracy on iWildCam and worst-region OOD accuracy on FMoW for supervised fine-tuning (FT). In "VIT in-context learning", we append 256 random patches, sampled from images within the same group, to the input patch sequence.

| | iWildCam | | | FMoW | | |
|---|---|---|---|---|---|---|
| | DINO ViT-S/8 | SWAG ViT-B/16 | CLIP ViT-L/14 | DINO ViT-S/8 | SWAG ViT-B/16 | CLIP ViT-L/14 |
| *Our implementations* | | | | | | |
| FT ViT | $75.7_{\pm 0.2}$ | $77.5_{\pm 0.5}$ | $81.5_{\pm 0.2}$ | $35.0_{\pm 0.2}$ | $39.8_{\pm 0.5}$ | $46.6_{\pm 1.1}$ |
| FT ViT in-context learning | $76.3_{\pm 0.6}$ | $77.6_{\pm 1.1}$ | $81.7_{\pm 0.4}$ | $35.1_{\pm 0.6}$ | $38.9_{\pm 0.7}$ | $46.3_{\pm 1.4}$ |
| FT ContextViT (w/o layerwise) | $77.5_{\pm 0.2}$ | $78.6_{\pm 0.4}$ | $81.9_{\pm 0.1}$ | $37.3_{\pm 0.6}$ | $41.3_{\pm 1.0}$ | $49.1_{\pm 0.7}$ |
| FT ContextViT | $\mathbf{77.7}_{\pm 0.3}$ | $\mathbf{79.6}_{\pm 0.6}$ | $\mathbf{82.9}_{\pm 0.3}$ | $\mathbf{37.7}_{\pm 0.5}$ | $\mathbf{41.4}_{\pm 0.3}$ | $\mathbf{49.9}_{\pm 0.4}$ |

Similar to the JUMP-CP dataset, the teacher network receives two global views of the image, while the student network receives six local views. Here, we provide an explanation of the data augmentation configurations for both the global and local views.

For the teacher network, the two global views are generated as follows: starting with the original pathology image, we first randomly crop the image with a scale sampled from the range of $[0.4, 1.0]$. Then, we resize the cropped image back to its original size of 96 by 96 using a bicubic transformation. A random horizontal flipping is applied with a probability of 0.5. Color jittering is performed with maximum brightness change of 0.4, maximum contrast change of 0.4, maximum saturation change of 0.4, and maximum hue change of 0.1, with a probability of 0.8. The image is transformed into grayscale with a probability of 0.2. Gaussian blur is applied using a kernel size of 1.0. Finally, solarization is applied to one of the global views with a threshold of 0.2.

For the student network, which receives six local views, a similar process is followed with the following changes: 1) The random cropping scale is adjusted to $[0.05, 0.4]$, ensuring that the student network observes only local views of the original image. 2) After cropping, the image is rescaled to a lower resolution of 32 by 32 instead of 96 by 96. 3) Gaussian blur is applied using a larger kernel size of 5. 4) Solarization is not applied to the local views.

## C.2 DINO PRE-TRAINING: OPTIMIZATION

For the Camelyon17 dataset, we employ the same optimization configuration as for the JUMP-CP dataset during DINO pre-training. This includes training with a batch size of 512 for 100 epochs, utilizing the AdamW optimizer with a cosine learning rate scheduler, and applying weight decay, among other techniques that we have discussed in Section B.2.

## C.3 LINEAR PROBING

To evaluate the resulting DINO embeddings, we utilize the standard linear probing test, which involves training a linear classifier while keeping the ViT backbones frozen. Consistent with Caron et al. (2021), we employ the following data augmentation and optimization methods:

For data augmentation, we first apply random cropping with a scale range of $[0.08, 1.0]$ and subsequently resize the resulting image to a resolution of 96 by 96. Additionally, we incorporate horizontal flipping with a probability of 0.5. Moreover, we apply color jittering with a probability of 0.8, where the maximum allowed changes include a brightness change of 0.4, a contrast change of 0.4, a saturation change of 0.4, and a hue change of 0.1.

In terms of optimization, we train the linear classifier for 100 epochs using a batch size of 512. Since the only trainable parameters are in the final linear layer, we do not employ warmup or weight decay techniques. To optimize the classifier, we employ SGD with a momentum value of 0.9. The learning rate is selected from the range $[0.0005, 0.001, 0.005]$, and we utilize a cosine annealing scheduler to decay the learning rate gradually throughout the training process.

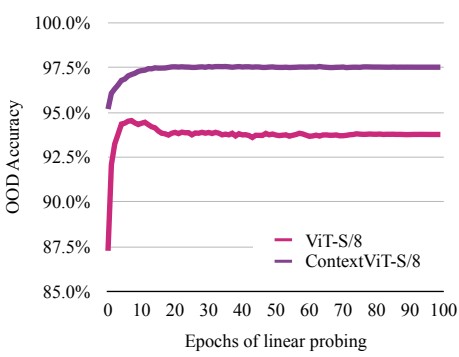

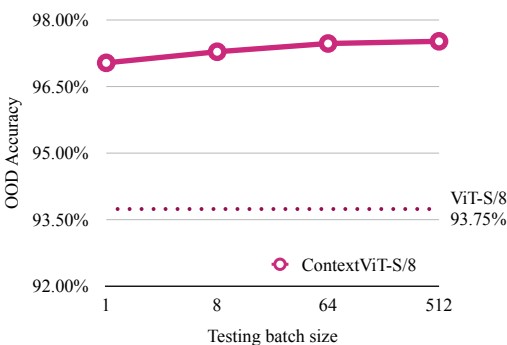

Figure 4: Learning curves of linear probing on Camelyon17. Unlike ViT-S/8, the OOD accuracy on top of ContextViT-S/8 steadily improves as the training of the linear classifier progresses.

Figure 5: On Camelyon17, we observe that even when we use a testing batch size of 1 (computing the context by aggregating 144 8 x 8 patches from a single testing image into a single context token), ContextViT-S/8 still significantly outperforms ViT-S/8.

# D    ADDITIONAL ANALYSIS

## D.1    SENSITIVITY TO TESTING BATCH SIZE

During the inference phase, ContextViT-S/8 takes a random batch of testing examples (512 in all previous experiments) and groups them based on their group membership, such as the hospital ID in the case of Camelyon17, to infer the corresponding context tokens. In Figure 5, we present a visualization of ContextViT's linear probing performance sensitivity across different testing batch sizes. Remarkably, even with a testing batch size of 1, where ContextViT-S/8 leverages patches from *a single testing image* (144 patches) to establish its context, it still outperforms our ViT-S/8 baseline by a significant margin (97.0 vs. 93.8). It's worth noting that after DINO pre-training, ContextViT-S/8 acquires the ability to condition its output features with respect to the context. As highlighted by Gao et al., pathology images from different hospitals exhibit distinct color distributions. The exceptional performance of ContextViT-S/8 with a small testing batch size demonstrates the model's capability to automatically estimate the color distribution shift from a few representative images during test time.

## D.2    PRACTICAL CONSIDERATIONS FOR BATCHES WITH MANY GROUPS

**Context-based data sampler**    In all of our experimental settings, we have utilized the standard random sampler, which selects examples uniformly at random from the dataset, for simplicity. However, when dealing with covariates that can take a large number of distinct values, an alternative approach to enhance the performance of direct mean pooling is to employ a context-based data sampler. By sampling a batch comprising examples with the same group membership, we can ensure an adequate number of instances for estimating the context token accurately. Adopting a context-based sampler also offers the advantage of eliminating the need for grouping examples during the forward pass, as the sampler guarantees that all samples within a batch belong to the same group. This can potentially lead to further improvements in computational efficiency. It is worth noting that to prevent divergence across different groups, gradient accumulation across batches may be necessary.

## D.3    EXPLORING ALTERNATIVE CONTEXT INFERENCE MODELS

The context inference model $h(\cdot; \phi)$ in our study serves as a mapping from a set to a vector representation. In the paper, we propose a simple approach that employs mean pooling followed by a linear transformation. However, there are other options available for parameterizing the context inference model. Here, we discuss additional approaches that can be considered.

ID images

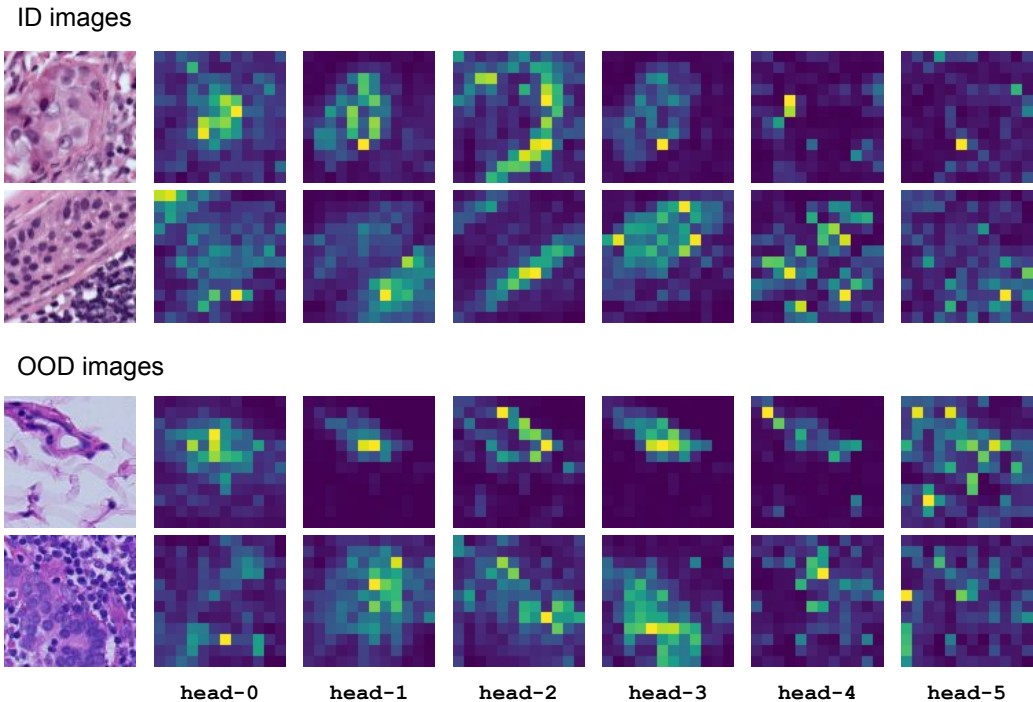

Figure 6: Attention heatmap of ContextViT on patholog images from in-distribution hospitals (top) and OOD hospitals (bottom) in Camelyon17.

**In-context learning by sampling patches from other images in the group**    As discussed in Section 3.1, one direct way to incorporate context conditioning is by appending image patches from other images within the same group into the patch sequence of the current image. However, this would result in an excessively long sequence length. Since self-attention scales quadratically (in terms of memory and compute) with respect to the sequence length, this approach is not feasible. To overcome this limitation, an alternative is to sample *a fixed number of patches* from these images and utilize them as the context. We explored this baseline approach by sampling 256 patches for the context and present its supervised fine-tuning results in Table 5. The results show mixed performance, likely due to the randomness in the context conditioning. It outperforms the ViT baseline on iWildCam but underperforms ViTs on FMoW. Furthermore, even this sampling approach significantly increases memory consumption compared to ContextViT. For instance, conditioning the CLIP model with 196 patches leads to a 49% increase in GPU memory size.

**Deep sets**    Deep sets Zaheer et al. (2017) offers a framework for dealing with objective functions defined on sets that are permutation-invariant. In our application, for each patch token embedding $t_{p_i}$, it utilizes an encoder network $\varphi(t_{p_i})$ to encode it. Then, it aggregates the embeddings of all patches belonging to the same group into a fixed-length vector using either sum pooling or max pooling. Finally, another network $\rho$ processes the aggregated representation to generate the final output.

In Section 3.4, we conducted experiments with a specific instance of the deep sets model, where we employed two multi-layer perceptrons (each with two hidden layers and ReLU activations) as the $\varphi$ and $\rho$ networks. Additionally, we incorporated residual connections for each hidden layer. We utilized the sum pooling mechanism to aggregate information across the set. As shown in Table 4, while this method exhibits increased representation power compared to mean pooling with a linear transformation, the latter demonstrates better generalization capabilities.

# E    VISUALIZATIONS

In this section, we present additional visualizations of ContextViT.

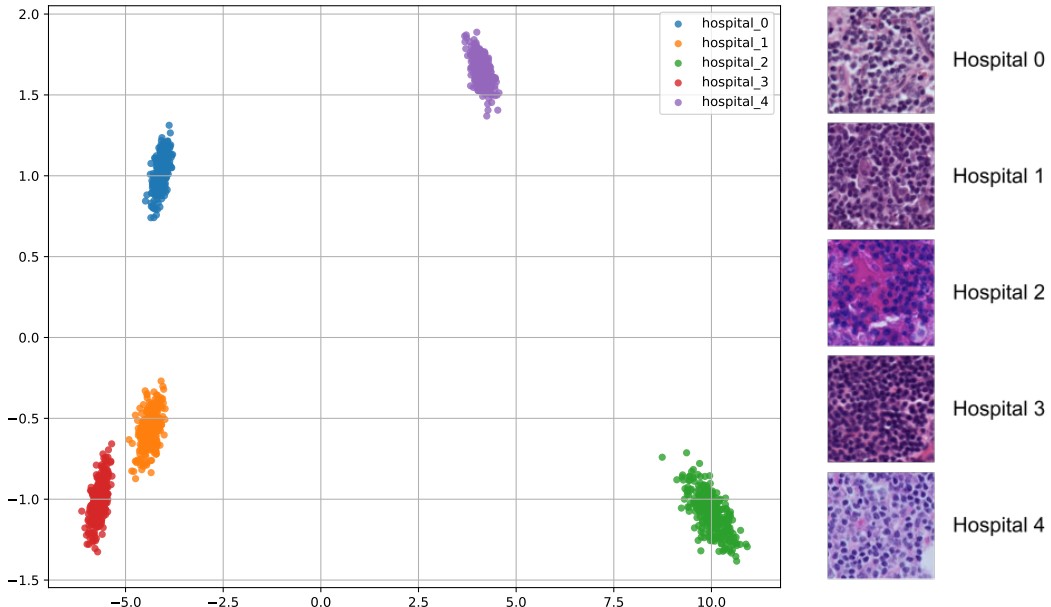

Figure 7: Left: PCA visualization of context tokens inferred by ContextViT using pathology images from different hospitals. Right: Example images from different hospitals.

**Attention maps from multiple heads**   In line with the findings of Caron et al. (2021), which demonstrate that DINO can learn class-specific features leading to unsupervised object segmentations, we visualize the attention heatmaps for different heads learned with ContextViT in the Camelyon17 dataset. Figure 6 illustrates these attention maps, showcasing that the learned attentions are focused on meaningful aspects for both in-distribution data and out-of-distribution data. Furthermore, different attention heads exhibit distinct preferences for different cell types. For instance, head-2 primarily focuses on the contour of the cells, while head-3 directs its attention to the actual cells themselves.

**Visualizing the context tokens**   We employ PCA to visualize the learned context tokens for each hospital, and the results are presented in Figure 7. One approach to visualizing the context tokens is by inferring them from all examples belonging to each hospital, resulting in five unique context tokens. However, in practice, we infer the context token on-the-fly for the current mini-batch. Using a batch size of 256, we sample 300 batches for each hospital.

Remarkably, the inferred context tokens for each hospital exhibit high similarity, appearing closely clustered together. Additionally, we include example images for each hospital on the right side of Figure 7. Notably, the distances between context tokens from different hospitals align well with their visual dissimilarities. For instance, hospital 3 (highlighted in red) is closer to hospital 1 (highlighted in orange) than it is to hospital 4 (highlighted in purple).

