# OpenReview forum: "Contextual Vision Transformers for Robust Representation Learning"
_ICLR.cc/2024/Conference — Submitted to ICLR 2024_

### Official Review · Reviewer_Y1Xv · 2023-10-30

**Soundness:** 4 excellent
**Presentation:** 4 excellent
**Contribution:** 3 good
**Rating:** 6
**Confidence:** 2

**Summary:**

The paper introduces Contextual Vision Transformers (ContextViT), a method to address structured variations and distribution shifts in image datasets. It leverages context tokens and token inference models to enable robust feature representation learning across groups with shared characteristics. The paper provides evidence of ContextViT's effectiveness through experiments in gene perturbation classification and pathology image classification.

**Strengths:**

- The paper introduces a novel method, ContextViT, to address structured variations and distribution shifts in image datasets. It brings a unique perspective to the problem of improving feature representations for vision transformers.
- The paper is well-written and provides clear explanations of the methodology, experiments, and results.
- ContextViT is extensively evaluated in different tasks, showcasing its effectiveness in improving out-of-distribution generalization and resilience to batch effects.

**Weaknesses:**

- How to chose and define the "in-context" prompt is unclear.
- While the paper is well-structured and well-written, it would be beneficial to include more detailed comparisons with related work to highlight the novelty of the proposed approach.
- In the "Out-of-Distribution Generalization (Pathology Images)" section, it's not entirely clear what "linear probing accuracy" means and how it relates to out-of-distribution generalization. A more in-depth explanation of this metric would improve the clarity of the paper.

**Questions:**

- Are there any specific use cases or domains where ContextViT is particularly well-suited, and are there any limitations or scenarios where it may not perform as effectively?
- Could the authors provide more insights into how ContextViT's approach to handling structured variations and distribution shifts could be applied in practical applications outside of the ones discussed in the paper?

---

> ### Author Response · Authors · 2023-11-22
>
> We thank the reviewer for their detailed review and constructive feedback.
>
> **Context Choice and Definition:** We carefully select the group membership variable, as depicted in Figure 2, ensuring datasets share a conditional dependency structure with their environment. For instance, pathology images can vary based on the staining procedures or machines used by different hospitals. To address this, we chose the hospital identity indicator as the context variable, which has shown to significantly improve model performance in our experiments.
>
> **Comparisons with Related Work:** Different from previous work, ContextViT explores token conditioning based on the group membership of the data. ContextViT introduces two novel technical enhancements:
> + Context inference network that maps example sets from the same group into a context token;
> + Layer-wise context conditioning that allows the ViT to integrate context at various network depths, enhancing generalization.
> + We also establish a principled mathematical link from ContextViT to in-context learning for distribution shifts.
>
> We have updated our related work section to better highlight these differences.
>
> **Linear probing:** Linear probing, a common protocol for evaluating self-supervised learning [1], involves training a linear classifier on top of fixed pre-trained ViT and ContextViT backbones to predict a target label, such as the presence of cancer. In the Cameylon17 dataset, the classifier is trained on images from three training hospitals and tested on a fourth, unseen hospital. Successful generalization to the held-out hospital indicates aligned feature distributions. We have expanded our discussion on this evaluation metric to provide further clarity.
>
> **Use Cases and Limitations:**
> + ContextViT is particularly well-suited for imaging applications where data are grouped or batched.
> + However, if the group membership variables are mis-specified and fail to capture the environmental context, ContextViT's performance may not exceed that of a regular ViT. For example, a randomly assigned context variable would not improve the model's performance. The effectiveness of ContextViT relies on the user's understanding of the dataset to accurately specify group memberships.
> + We appreciate the reviewer's insight on this matter and have included a discussion on the potential complexities and dependencies that could affect generalization in our paper.
>
> **Practical Applications and Structured Variations:**
> + This links very well to the previous question. In practice, users want to consider invariant factors of variation per group that systematically shift across groups. For example, shared environments over datasets or global variables in the plated context (see Figure 2 for a graphical model interpretation).
> + We also note that oftentimes this can be an exploratory process where people need to train specific models with different contexts and see whether the resulting models explain away the distributions shifts when there is no obvious choice of the context variable.
> + Lastly, the question interfaces with our desire for a more formal framework in future work based on statistical tests on top of ViT embeddings to detect biases over various covariates and suggest useful context,  which would be a useful tool and would link this work more to causal inference.
>
> We trust that this rebuttal has addressed the points raised by the reviewer. We are thankful for your time and effort and are open to any further questions.
>
> **References:**
> 1. Chen, Ting, et al. "A simple framework for contrastive learning of visual representations." International conference on machine learning. PMLR, 2020.

---

> > ### Comment · Reviewer_Y1Xv · 2023-12-05
> >
> > Thanks authors for the detailed responses, which well solve my concerns. However, I agree with the other reviewers about the need for greater novelty in the proposed approach.

---

### Official Review · Reviewer_hyoq · 2023-10-30

**Soundness:** 3 good
**Presentation:** 3 good
**Contribution:** 3 good
**Rating:** 5
**Confidence:** 4

**Summary:**

This work proposes a Contextual Vision Transformers (ContextViT) based on ViT. ContextViT is designed for adapting ViTs to OOD data with varying latent factors. This work is inspired by in-context learning and prepends tokens to input sequences for alleviating model performance. This paper finds out that standard context tokens might not be able to generalize to unseen domains, therefore it proposes a context inference network that estimates context tokens from input images. The proposed method is evaluated with cell-imaging and histopathology datasets and achieves performance improvements under distribution shifts.

Pros:

- This paper is well-written and easy to follow.
- Figure 1 is well drawn to illustrate the overall idea of this work.
- Layer-wise context conditioning is well-motivated and makes sense.

Cons:

The novelty of this work is limited.
- The intrinsic difference between this work and visual prompting [1] is unclear. It seems that visual prompting can also fit this OOD scenario.
- The key idea of this work is similar to [2], which also uses a network to predict the context/domain tokens.
- The comparison in experiment section is insufficient.
- Lack of visualization of the learned context token, which shows the difference of context tokens of different groups.

The paper is simple and effective, but its novelty is unfortunately limited, and analysis for the insight of this approach is absent.

[1] Jia, Menglin, et al. "Visual prompt tuning." European Conference on Computer Vision. Cham: Springer Nature Switzerland, 2022.
[2] Zhang, Xin, et al. "Domain Prompt Learning for Efficiently Adapting CLIP to Unseen Domains." arXiv preprint arXiv:2111.12853 (2021).

**Strengths:**

- This paper is well-written and easy to follow.
- Figure 1 is well drawn to illustrate the overall idea of this work.
- Layer-wise context conditioning is well-motivated and makes sense.

**Weaknesses:**

The novelty of this work is limited.
- The intrinsic difference between this work and visual prompting [1] is unclear. It seems that visual prompting can also fit this OOD scenario.
- The key idea of this work is similar to [2], which also uses a network to predict the context/domain tokens.
- The comparison in experiment section is insufficient.
- Lack of visualization of the learned context token, which shows the difference of context tokens of different groups.

The paper is simple and effective, but its novelty is unfortunately limited, and analysis for the insight of this approach is absent.

**Questions:**

-

---

> ### Author Response · Authors · 2023-11-22
>
> We thank the reviewer for their detailed review and constructive feedback.
>
> **Related work:** We are grateful for the additional related work highlighted by the reviewer. We have incorporated these references into our paper to provide a more comprehensive context for our contributions.
> + Visual prompt tuning: Unlike visual prompt tuning, which learns extra tokens for each downstream task, our work defines the extra token based on the group membership of the data. This distinction is crucial for enabling our model to adapt to new distributions without task-specific tokens.
> + Domain prompt learning: Similar to visual prompt tuning, domain prompt learning generates a consistent output prompt for each task, as the text label set remains unchanged. In our work, we don’t consider text inputs as they are not meaningful in cross-distribution generalization. For instance, the CLIP text encoder would not produce meaningful representations using the group name such as  “plate BR00116991” and “plate BR00116993” or “hospital A” and “hospital B”. Moreover, ContextViT's extra token is dynamically derived from the group membership, allowing for generalization beyond the scope of fixed tasks.
>
> **Novelty:** In addition to our unique perspective on token conditioning based on the data groups, ContextViT is complemented by two novel technical enhancements:
> + Context inference network maps sets of examples from the same group into a context token, enabling ContextViT to generalize to unseen groups during training.
> + Layer-wise context conditioning enables the ViT to integrate context at various depths within the network, not just at the input layer, which significantly improves generalization.
> + Finally, we provide a principled mathematical linking from ContextViT to in-context learning for distribution shift.
>
> **Comparison in the experiment section:**
> + ContextViT has been evaluated as a plug-and-play enhancement for three established pre-trained ViTs (DINO, SWAG, CLIP) and has consistently outperformed recent fine-tuning baselines in generalization tasks.
> + In self-supervised representation learning, ContextViT has set a new state-of-the-art on the Camelyon17-WILDS dataset, outperforming 26 other baselines. The full comparison is available on the WILDS leaderboard (https://wilds.stanford.edu/leaderboard).
>
> **Analysis of the model:**
> + We have conducted a thorough comparison of ContextViT with its non-context-conditioned ViT counterparts (Tables 1, 2, and 3).
> + Our ablation studies and runtime comparisons (Table 4) carefully examine the impact of layer-wise context conditioning and various implementations of the context inference network.
> + Figures 5 (in the appendix)  explores ContextViT's effectiveness with limited data for context token inference.
> + Figure 7 (in the appendix)  provides a visualization of the context token. We observe that the similarity between the context tokens are closely related to the similarity of the input images.
> + Figure 6 (in the appendix) provides a visual representation of the attention heatmap produced by ContextViT, illustrating its interpretability.
>
> We trust that this rebuttal has addressed the points raised by the reviewer. We are thankful for your time and effort and are open to any further questions.

---

> > ### Comment · Reviewer_hyoq · 2023-12-05
> > **Response**
> >
> > Dear authors.
> >
> > Thanks for your effort in rebuttal. This paper has some merits and achieved promising experiment results. However the novelty is not enough to be accepted as an ICLR paper. So I maintain my score.

---

### Official Review · Reviewer_4Dw1 · 2023-11-01

**Soundness:** 3 good
**Presentation:** 3 good
**Contribution:** 2 fair
**Rating:** 5
**Confidence:** 3

**Summary:**

The paper proposes ContextViT to address the distribution shift between different datasets. ContextViT uses a context inference model taking the dataset as input to get a context embedding for the dataset, and predicts the label conditioned on the context embedding (token). It also makes this process layer-wise to capture different-scale distribution shift.

**Strengths:**

The paper presents a method to mitigate the distribution gap between different datasets. Based on their experimental results, the proposed method, ContextViT, has the ability to improve the performance under distribution shift.

**Weaknesses:**

- The paper mentioned that the proposed method applies the concept of in-context learning in vision transformer. However, in my opinion, in-context learning is a kind of few-shot learning, which predicts based on the (data, label) pair of a few samples, unlike the usage of all the dataset-c data (or a batch of the data) in this paper. The method looks like a summarization of the dataset information and then makes the prediction based on that summarization.

- The method requires a lot of distribution-c data at the inference stage and increases the inference overhead.

- The oracle-context model is very similar to some prompt tuning works, like Visual Prompt Tuning & Prompt Learning for Vision-Language Models, but these works are not discussed in the paper.

**Questions:**

Please see weaknesses.

---

> ### Author Response · Authors · 2023-11-22
>
> We thank the reviewer for their detailed review and constructive feedback.
>
> **Concept of in-context learning:** We appreciate your input on this matter. To clarify, we have updated our methods section to distinguish between data conditioning and data+label conditioning. In-context learning is a wide catch term for various types of data conditioning. We consider the case of in-context learning for generalization under group membership of the data and derive our method under that assumption. This is for instance when we condition the model on different environments, a very common assumption. This consideration is also driven by the reality that labels for test distributions are often unavailable, whereas data is plentiful. The context inference network is designed to summarize dataset information, enabling the vision transformer to adapt its representation of the input image accordingly.
>
> **Requires a lot of distribution-c data at the inference stage:**
> + While in an ideal world we would need all the distribution-c data to derive the context token, we empirically tested the data hunger of the model. In Figure 5, we demonstrate ContextViT's performance across a range of testing batch sizes, from 1 to 512. The model's ability to infer context improves with more examples; however, it still achieves state-of-the-art performance with as few as 8 examples, outperforming all existing benchmarks on the WILDS leaderboard (https://wilds.stanford.edu/leaderboard).
> + We also highlight that in many real-world settings, such as new hospitals, devices, users, or biological experiments, testing data is naturally batched. ContextViT is well-suited for these practical applications.
>
> **Related work:** We are grateful for the identification of missing references. These have now been incorporated into our revised paper. Our approach is distinct in its use of group membership to define the context token, coupled with two innovative technical contributions:
> + Context inference network maps sets of examples from the same group into a context token, allowing ContextViT to generalize to new groups unseen during training.
> + Layer-wise context conditioning enables the ViT to integrate context at various network depths, not just at the input layer, which significantly improves generalization.
> + Finally, we provide a principled mathematical linking from ContextViT to in-context learning for distribution shift.
>
> We believe this rebuttal has addressed the points raised by the reviewer. We are thankful for your time and effort and are open to any further questions.

---

### Official Review · Reviewer_kTos · 2023-11-01

**Soundness:** 2 fair
**Presentation:** 2 fair
**Contribution:** 2 fair
**Rating:** 5
**Confidence:** 4

**Summary:**

This paper introduces an improved ViT where some group-specific context information from the sub-groups in datasets is collected and generated from those images in a group. The network generates the context token from those images and appends them to image patch embeddings. Their experiments show some improvement of this ViT on some group-specific datasets.

**Strengths:**

The idea of capturing context information from the datasets is interesting.
The writing of this method is clear and easy to follow.
The experiments demonstrate the efficiency of their proposed framework on both the dataset with the same distribution and other datasets with different distributions.

**Weaknesses:**

The view and impact of this paper are limited. It seems the method focuses on improving the performance of the datasets that contain several distinct groups. Although the authors demonstrate improvements on some specific datasets, the improvement in general image tasks is still unclear. It is suggested to widely evaluate their framework on other popular datasets and tasks or extend related techniques to improve the capability of transfer learning from one task to some other tasks. It should also be compared with more related works.


Despite the proposed contextual learning paradigm, the technical contributions in this paper are limited and not novel enough.


Some unclear presentations:

1. Figure 1 is unclear and somehow misleading. The source of the context (where those images come from) and the function (input, output) of the inference model should be labeled. I strongly suggest redoing this figure.

2. The end of page 5 is missing.

3. Table 2 looks messy and should be redesigned.

**Questions:**

What would the performance be if we want to apply this framework to a large dataset that was combined with several small datasets?

If we don't know the sub groups of the data, is there anyway to benefit from the proposed framework?

---

> ### Author Response · Authors · 2023-11-22
>
> We thank the reviewer for their detailed review and constructive feedback.
>
> **Scope of the paper:** We agree that ContextViT is tailored for scenarios with grouped datasets, which is a common occurrence in real-world applications where data often arrives in batches or groups. The focus of this paper is on enhancing robustness across distribution shifts. We show that instead of having to learn one model per distribution, we can share the parameters across the tasks and utilize the context inference network to inform the model of the specific distribution. We separately show that by doing so and learning a context inference network we can also learn to generalize to new test distributions effectively.
>
> **Improvement in general image tasks:**
> + In this paper, we have included three widely-recognized imaging benchmarks from WILDS and a recently released cell imaging benchmark released by the Broad Institute.
> + The reviewer asks us to focus on general image tasks. We highlight that instead of going for artificial benchmarks, we went for real world datasets that exhibit specific problem structure that is not tackled by more general machinery. We would ask the reviewer to see the value of overcoming unsolved problems rather than iterating on known problems.
> + To be more specific, in many practical applications (especially scientific domains such as biological discovery), being able to deal with batch effects is a key hindrance to the successful application of representation learning. This is exemplified by our results on the recent JUMP-CP cell painting dataset released by the Broad institute of Havard and MIT, for which ContextViT consistently improves the generalization across different data distributions. We also extensively tested ContextViT on pathology imaging such as Camelyon17-WILD. This is a key problem as real world applications like medical imaging often suffer from distribution shifts in batches due to the employment of different staining procedures or machines across hospitals. We hope we have demonstrated the importance of the task that we tackle to the reviewer.
>
> **Evaluation and baselines:**
> + ContextViT has been tested as a plug-and-play enhancement for three well-established pre-trained ViTs (DINO, SWAG, CLIP) in a supervised learning context. It has consistently outperformed two recent fine-tuning baselines in terms of generalization.
> + In the realm of self-supervised representation learning, we have considered both the in-distribution generalization and out-of-distribution generalization. In the first, ContextViT consistently outperformed ViT across all testing configurations. In the second, ContextViT has achieved a new state-of-the-art on the Camelyon17-WILDS dataset, surpassing 26 other baselines. Due to space constraints, we only included the top baselines in Table 3. The complete list of baselines is available on the WILDS leaderboard (https://wilds.stanford.edu/leaderboard).
>
> **Technical contributions:** We recognize prior work on conditioning ViTs with extra tokens in our related work section. Our approach uniquely leverages group membership to define the context token and introduces two novel technical enhancements:
> + The context inference network effectively maps sets of examples from the same group into a context token. This enables ContextViT to generalize on new groups that it hasn’t seen during training, as demonstrated by our empirical studies.
> + Layer-wise context conditioning allows the ViT to utilize context at multiple levels of the network, rather than just the input layer, enhancing generalization capabilities.
> + Finally, we provide a principled mathematical linking from ContextViT to in-context learning for distribution shift.
>
> **Clarity:**
>   + Based on your suggestions, we have revised the figure to enhance its clarity and readability.
>   + The end of page 5 continues on page 6 (under Table 1).
>
> **Application to large datasets with small sub datasets:** The JUMP-CP benchmark, comprising multiple data plates from various perturbations, serves as an example of combining "small datasets".
>
> **Without knowledge of the group membership, is there any way to benefit from the proposed framework?**
> + For this paper, we assume that the indicator function of the group membership is known during training, a realistic assumption given that data is often generated in group fashion in the real world. In future work, one might strive to infer latent group membership from the data, but we don’t tackle that in this submission.
> + In Figure 5 (in the appendix), we explored the performance of ContextViT with a testing batch size of one. This would be equivalent to the situations where we don’t assume knowledge of the group membership at test time. We have demonstrated that this self-conditioning still improves the performance on top of the standard ViT.
>
> We trust that this rebuttal addresses the concerns raised. We are grateful for your time and effort and welcome any further questions.

---

> ### Comment · Reviewer_kTos · 2023-12-05
> **Response**
>
> Dear Authors,
>
>
> Thank you for your effort and revisions. However, I remain unconvinced about the broader impact of your method. Please provide more substantial evidence of its applicability to general image recognition tasks. Additionally, I agree with the other reviewers about the need for greater novelty in your approach. As it stands, I maintain my score and believe the paper is currently below the acceptance threshold.
>
>
> Best regards,
>
> Reviewer

---

### Author Response · Authors · 2023-11-22

We would like to extend our sincere thanks to the reviewers for their thorough evaluations and constructive feedback. In light of their comments, we have carefully revised our manuscript and would like to summarize the pivotal contributions of our study, as well as the enhancements made during the rebuttal period.

**Key Contributions of Our Work:**
+ We introduce ContextViT, an novel adaptation of ViT that generates robust image representations. ContextViT leverages learned context tokens to effectively capture and adapt to distribution shifts, drawing inspiration from in-context learning principles which we derived mathematically in the paper.
+ We have embedded a context inference mechanism within ContextViT, enabling it to adapt and generalize to novel, unseen distributions in real-time.
+ We propose a layer-wise context conditioning approach for ContextViT, which employs per-layer context tokens in a sequential manner throughout the transformer layers. This technique is designed to address concept-level variations across different distributions.
+ Through rigorous evaluation in both supervised fine-tuning and self-supervised learning scenarios, ContextViT has demonstrated marked performance improvements when faced with distribution shifts, thereby setting a new benchmark for state-of-the-art performance. Our code will be released after the submission.

**Manuscript Updates:**
+ Figure 1 has been revised for enhanced clarity in depicting the model's architecture.
+ In Section 3.1, we have elaborated on the distinctions between in-context conditioning using data-label pairs and conditioning solely based on data.
+ Section 4.3 now includes a more detailed explanation of the linear probing process and its implications for out-of-distribution generalization.
+ A new paragraph has been added to Section 4.3 to address practical considerations when implementing ContextViT in real-world scenarios.
+ Section 5 has been updated to incorporate the references suggested by the reviewers. We have also provided a comparative discussion of ContextViT against these prior works.

We hope we have demonstrated to the reviewers that our method is principled and tackled the edges of representation learning necessary to achieve real world impact in many typically overseen tasks dealing with collections of small or diverse datasets, which are typically seen in scientific applications or sensing.

---

### Meta-Review · Area_Chair_xUPB · 2023-12-09

**Metareview:**

Paper proposes an approach, named Contextual Vision Transformers (ContextViT), which is designed to generate image representations for datasets with distributional shifts in latent factors across various groups. ContextViT leverages context tokens to encapsulate group-specific information. The paper was reviewed by 4 reviewers and received: 3 x Marginally below the acceptance threshold and 1 x marginally above the acceptance threshold ratings.

The main concerns raised with the paper centered around novelty and ability to distinguish contributions from visual prompting (which in the very least is closely related). Authors attempted to address these and other concerns through the rebuttal, but their arguments were not particularly convincing to the reviewers. In discussion, a consensus of the paper not being good enough has emerged. Even the most favorable reviewer [Y1Xv] agrees that novelty is lacking.

AC has read the reviews, rebuttal and the discussion that followed and agrees with the consensus of reviewers that contributions of the work maybe limited and the similarity to visual prompting (and comparison) to such methods is necessary to better characterize the contributions. Therefore the decision is to reject the paper at this time.

**Justification For Why Not Higher Score:**

The approach does appear to be a form of prompting, while not clearly positioning as such. From the pure technical and significance perspective it also appears to be lacking. Given that there is reviewer consensus for Rejection, I do not see bases for overturning such a decision.

**Justification For Why Not Lower Score:**

N/A

---

### Decision · Program_Chairs · 2024-01-16

Reject